# A Review of Advanced Impedance Biosensors with Microfluidic Chips for Single-Cell Analysis

**DOI:** 10.3390/bios11110412

**Published:** 2021-10-22

**Authors:** Soojung Kim, Hyerin Song, Heesang Ahn, Taeyeon Kim, Jihyun Jung, Soo Kyung Cho, Dong-Myeong Shin, Jong-ryul Choi, Yoon-Hwae Hwang, Kyujung Kim

**Affiliations:** 1Departments of Congo-Mechatronics Engineering, Pusan National University, Busan 46241, Korea; kimsoo9640@gmail.com (S.K.); rin8520@gmail.com (H.S.); ahn3890@pusan.ac.kr (H.A.); ktyeon09@gmail.com (T.K.); jhyun4210@gmail.com (J.J.); 2Division of Nano Convergence Technology, Pusan National University (PNU), Miryang 50463, Korea; sookyungcho@pusan.ac.kr; 3Department of Mechanical Engineering, The University of Hong Kong, Pokfulam, Hong Kong 999077, China; dmshin@hku.hk; 4Medical Device Development Center, Daegu-Gyeongbuk Medical Innovation Foundation (DGMIF), Daegu 41061, Korea; jongryul32@dgmif.re.kr; 5Department of Nano Energy Engineering, Pusan National University (PNU), Busan 46241, Korea; 6Department of Optics and Mechatronics Engineering, Pusan National University, Busan 46241, Korea

**Keywords:** impedance sensor, microfluidic chip, biosensor, single cell trapping

## Abstract

Electrical impedance biosensors combined with microfluidic devices can be used to analyze fundamental biological processes for high-throughput analysis at the single-cell scale. These specialized analytical tools can determine the effectiveness and toxicity of drugs with high sensitivity and demonstrate biological functions on a single-cell scale. Because the various parameters of the cells can be measured depending on methods of single-cell trapping, technological development ultimately determine the efficiency and performance of the sensors. Identifying the latest trends in single-cell trapping technologies afford opportunities such as new structural design and combination with other technologies. This will lead to more advanced applications towards improving measurement sensitivity to the desired target. In this review, we examined the basic principles of impedance sensors and their applications in various biological fields. In the next step, we introduced the latest trend of microfluidic chip technology for trapping single cells and summarized the important findings on the characteristics of single cells in impedance biosensor systems that successfully trapped single cells. This is expected to be used as a leading technology in cell biology, pathology, and pharmacological fields, promoting the further understanding of complex functions and mechanisms within individual cells with numerous data sampling and accurate analysis capabilities.

## 1. Introduction

Biosensors are defined as devices and measurement systems that include probes capable of detecting selected biological materials [1]. In general, biosensors use physical, optical, electrical, or chemical modalities to accurately measure changes in specific biological materials. Biosensors are typically categorized in terms of their measurement modalities; among them, electrical biosensors have the advantages of being label-free to acquire biological information, high-sensitivity, an accessibility of miniaturization of biosensors by fabrication processes to form small-sized electrical devices, and high cost-effectiveness. Electrical biosensors are subdivided into potentiometric, amperometric, and impedance sensors, depending on which electrical information is measured to obtain biological information [2,3]. These electrical biosensing techniques have been employed in devices and systems that detect and monitor various biological materials in a range from tissues to small biomolecules such as deoxyribonucleic acid (DNA) [4,5,6,7,8]. Specially, impedance biosensors have been used to detect biological components and molecules by measuring impedance changes without requiring special labeling processing, including cells, DNA, and proteins. Impedance biosensors apply sinusoidal voltage to specific frequencies and measure electrical impedance with alternating current flow [9].

The sinusoidal AC signal flows through the electrode, measures the amplitude and phase change of the sine wave response signal through the target material, and measures the composite impedance. This approach has been used to study the electrochemical phenomena of target materials over a wide range of frequencies. Research has been conducted to improve sensitivity by creating a variety of electrode structures, and to improve selectivity by coating the electrodes with the target material acceptor. The advantage is that electrical properties can be identified quickly and easily without the use of complex and noise-causing labeling techniques.

Microfluidics are techniques for devices and systems that can control small-scale fluids, and various microfluidic devices have been actively applied in that they can simulate and analyze specific biological activities in small-sized devices using a small amount of specimen [10]. As a representative example, cell sorting, counting, and trapping miniaturized using microfluids have been developed as tools to replace existing research equipment at low cost [11,12,13,14,15,16]. In addition, several research groups have developed microfluidic cell culture devices that mimic human organs to explore specific cellular phenomena or to analyze the effect or toxicity of drugs [17,18,19,20]. One of the advantages of microfluidic devices is their ability to install various analytical modalities; therefore, these microfluidic biosensors have been expanded in applications ranging from analytical tools in research to healthcare and industry [21,22,23].

In this article, we explored the concept of and recent studies on microfluidic impedance single-cell biosensors, a device that combines an electrical impedance biosensor and a microfluidic device to perform analysis on a single cell. First, we explored the definition, principle, and application of impedance microfluidic sensors in various biological applications to inform useful technology to be used as a tool for single-cell analysis. Second, concepts and recent studies of microfluidic devices that can classify single cells and move them to specific locations for impedance single-cell biosensing were also investigated and presented. Finally, we sequentially analyzed recent research on microfluidic impedance biosensors for single cells. The results of the investigation and analysis of microfluidic impedance sensors for single-cell analysis are expected to present a milestone in the development of advanced biosensing systems that analyze the characteristics of single cells using microfluidic impedance sensing.

## 2. Various Applications of Microfluidic Impedance Biosensors

Impedance sensors have the advantage of being able to detect minute changes in substances on working electrodes through measurements of changes in electrical impedance. Using this characteristic, research on impedance biosensors that can detect information (for instance, concentrations) of selected biomaterials in small amounts of specimens has been actively conducted. In particular, several studies have been conducted to combine microfluidic devices with impedance biosensors to reduce both the number of used samples and the overall size of the sensing system. In this chapter, we will explore the research and applications of these microfluidic impedance biosensors.

A general method for detecting a specific virus is real-time polymerase chain reaction (PCR). This method is highly accurate, but it takes a long time for the diagnostic results to be obtained. Microfluidic impedance virus sensors have been actively studied because they provide the detection of selected viruses, which are simpler and faster than the conventional method. R. Wang et al. developed a technique that can test avian influenza virus H5N2 with high sensitivity using a portable impedance sensor, which consists of 25 pairs of microelectrodes and a microfluidic channel, and antibody-coated magnetic nanobeads [24]. For performance comparison between the microfluidic impedance sensor and real-time reverse transcriptase PCR, there were two parameters, a sensitivity and specificity, used as performance indicators. A sensitivity was defined as a ratio of a number of samples identified as positive by both a virus analytical method (impedance sensor/real-time reverse transcriptase PCR) and virus culture (N_TP_) to a sum of N_TP_ and a number of samples identified as negative by the analytical method and positive by virus culture (N_FN_). Also, a specificity was calculated as a ratio of a number of samples identified as negative by both the analytical method and virus culture (N_TN_) to a sum of N_TN_ and a number of samples identified as positive by the analytical method and negative by virus culture (N_FP_). For viral samples extracted from the trachea of infected or normal chickens, the developed microfluidic impedance sensor provided 100% sensitivity (equal to the sensitivity of real-time reverse transcriptase PCR) and 64% specificity (less than the specificity (=100%) of real-time reversed transcriptase PCR). In the case of samples extracted from cloacal swabs of infected or normal chickens, the microfluidic impedance sensor had an ideal specificity (100%), which was higher than it (=69%) of real-time reversed transcriptase PCR, and lower sensitivity (55%) than real-time reversed transcriptase PCR-based virus detection (81%). Compared to PCR, the developed microfluidic impedance virus sensor has advantages of being small in size (portable), having shorter measurement time (30 to 60 min), and being easier to apply. Compared to the virus detection method using real-time reverse transcriptase PCR, the microfluidic impedance sensor provided 100% sensitivity and 64% specificity (less than real-time reverse transcriptase PCR) for viral samples extracted from the trachea of infected or normal chickens. For H5N2 samples extracted from cloacal swabs of infected or normal chickens, the developed microfluidic impedance sensor had an ideal specificity (100%) and lower sensitivity (55%) than real-time reverse transcriptase PCR-based virus detection (81%). Based on the separation of viruses using antibody-coated magnetic microbeads and additional biological labeling using chicken red blood cells (RBCs), J. Lum et al. developed a microfluidic impedance avian influenza virus H5N1 sensor with highly improved sensitivity and specificity [25]. R. Singh et al. developed microelectrodes coated with reduced graphene oxide (RGO) and antibodies corresponding to influenza virus H5N1 and established a microfluidic impedance influenza virus detection system where these electrodes were integrated (Figure 1a) [26]. Microelectrodes enhanced by RGO and their based influenza virus biosensors provided a limit of detection of 0.5 pfu/mL. The separation of viruses using magnetic nanoparticles coated with antibodies and electrical impedance virus sensing are also used to develop rapid diagnostic devices for human immunodeficiency virus (HIV-1) [27]. The microfluidic impedance HIV-1 biosensing device developed by H. Shafiee et al. presented the possibility of providing virus detection and point-of-care (POC) diagnosis more quickly than conventional analytical methods such as enzyme-linked immunosorbent assay (ELISA). To make this technique more suitable for use in POC diagnostics, X. Li et al. fabricated working electrodes coated with carbon ink, zinc oxide (ZnO) nanowires, and antibodies that can be attached to p24 antigen, one of the HIV’s biomarkers on a paper fluidic substrate [28]. They applied the electrodes to high-sensitive HIV detection using electrical impedance spectroscopy (A limit of detection of p24 antigen = 0.4 pg/mL). Also, this device and detection technique demonstrated the potential to be utilized for rapid diagnosis of COVID-19 by serological analysis. Compared to ELISA, microfluidic impedance sensor to detect virus enable faster detection and the use of low-cost antibody reagents since analytes are placed much closer to the detector. Also, electrical signal detection in the microfluidic impedance sensor can be more sensitive than optical measurement in ELISA.

Techniques for microfluidic impedance sensors that detect deoxyribonucleic acid (DNA) or protein biomarkers with high sensitivity and selectivity for early disease detections have been actively conducted to meet the growing demand for early diagnosis and treatment. Javanmard and Davis investigated an impedance biosensor to detect the hybridization of the target DNA [29]. This impedance sensor was equipped with a functionalized microfluidic pore by probe DNA and electrodes to measure the conductivity, so when a bead functionalized by target DNA passed, it attached to the microfluidic pore and DNA hybridization could be detected by a change in impedance. Ben-Yoav et al. established a miniaturized microfluidic impedance sensing platform with a size of 3.5 × 3.5 cm^2^ for the detection of DNA hybridization [30]. The platform was theoretically capable of measuring hybridization of target DNA with concentrations of 3.8 nM, and verification of actual experiments confirmed that DNA hybridization of target DNA with concentrations of 0.01 to 10 μM can be measured by the developed microfluidic impedance DNA hybridization sensor. In addition, studies have been conducted to detect DNA biomarkers of specific diseases using microfluidic impedance sensors for rapid diagnosis of the disease. As a representative study, Pursey et al. investigated a microfluidic multimodal sensor combined with electrochemical impedance sensing and optical surface plasmon resonance sensing to detect a DNA biomarker of bladder cancer, as described in Figure 1b [31]. In impedance-based bladder cancer biomarker detection, the microfluidic sensor provided a dynamic range that could detect DNA biomarkers at concentrations of 100 nM to 200 fM. Teengam et al. developed a paper fluidic device with integrated electrodes for electrochemical impedance measurement-based Mycobacterium tuberculosis detection [32]. Also, Alsabbagh et al. established a miniaturized microfluidic impedance sensing device with low sensitivity to human serum albumin, a non-specific biomolecule, and high sensitivity to troponin I, a biomarker of cardiovascular diseases [33].

Several research groups have studied on microfluidic impedance sensors to detect dangerous ingredients such as bacteria in food and pesticide residues as an inspection device that can test foods in the field. Tan et al. developed a microfluidic impedance sensor to detect O157:H7 and *Staphylococcus aureus* [34]. An antibody-functionalized nanoporous membrane was employed in the microfluidic impedance sensor to capture specific bacteria. Dastider et al. established a microfluidic impedance O157:H7 sensing device that can measure very low concentrations of bacteria from specimens using an array of ramped down focusing electrodes integrated by microelectromechanical system (MEMS) fabrication [35]. The impedance sensing platform, which enabled detection of O157:H7 bacteria at concentrations of 39 CFU/mL within two hours, demonstrated the availability of rapid sample testing. In addition, a microfluidic impedance sensor was developed to detect Salmonella in food samples as described in Figure 1c [36]. The impedance sensor, including micro-gaped integrated electrodes, is capable of testing Salmonella at concentrations of 300 cells/mL within an hour. Guo et al. investigated a polydimethylsiloxane (PDMS) miniaturized microfluidic impedance sensor for the detection of pesticide residues [37]. This impedance sensor can detect chlorpyrifos, an organophosphate pesticide, in a specimen from real vegetables such as leek, lettuce, and cabbage.

**Figure 1 biosensors-11-00412-f001:**
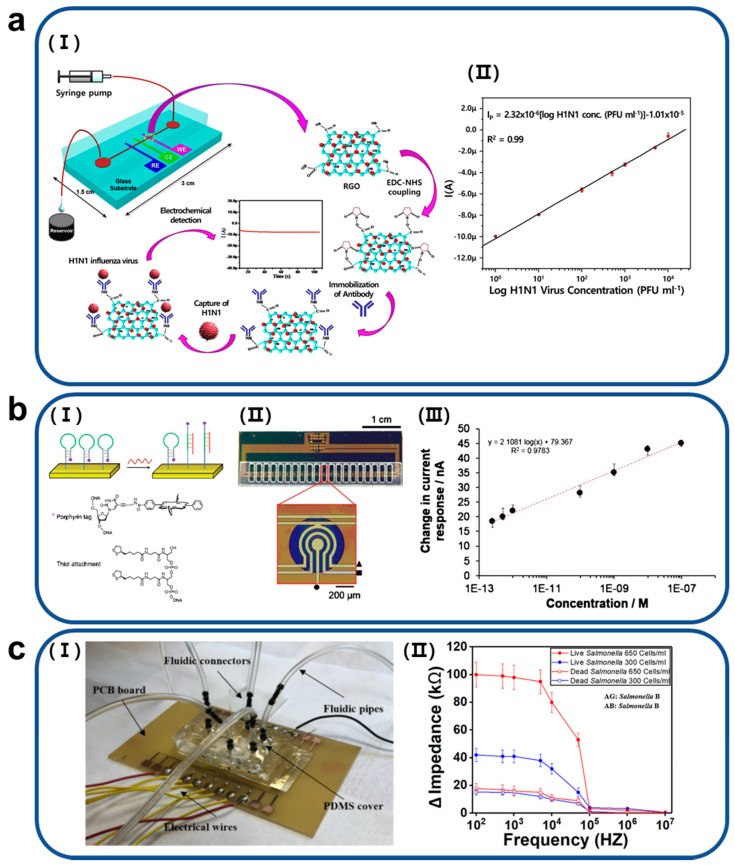
(**a**) (I) A schematic of a microfluidic impedance sensor to detect avian influenza virus H5N1 using microelectrodes coated by reduced graphene oxide (RGO). (II) Calibration plot presenting injected H1N1 virus concentrations (from 1 to 104 PFU mL^−1^) and the amperometric current through the electrochemical immunosensor with microfluidics. Reproduced with permission from [26] Copyright Scientific Reports 2017. (**b**) (I) A schematic of principle of an implementation of probe (hairpin) DNA to detect a biomarker of bladder cancer and an array of impedance sensing electrodes integrated in a microfluidic bladder cancer biomarker sensing device. (II) A graph of the relationship between biomarker concentrations and electrical current variations. The result indicates that the developed microfluidic impedance sensor allows linear measurements of DNA biomarkers of bladder cancer at concentrations of 100 nM to 200 fM. Reproduced with permission from [31] Copyright Sensors and Actuator B: Chemical 2017. (**c**) (I) A fabricated microfluidic impedance sensor to detect Salmonella from food samples and (II) a result of impedance spectroscopic measurements of live or dead Salmonella at two different concentrations (300 or 600 cells/mL) using the microfluidic impendence sensor. Reproduced with permission from [36]. Copyright PLOS ONE 2019.

In addition, microfluidic impedance sensing devices have been employed as tools to analyze specific cell characteristics under in vitro conditions. As a representative study, Nguyen et al. developed a microfluidic electrical impedance sensor that can accurately measure the migration of cancer cells cultured in a three-dimensional extracellular matrix and analyzed the metastasis of cancer cells [38]. This microfluidic impedance sensor has the advantage of being able to monitor cell migration at the single-cell level in real time. Pan et al. investigated a microfluidic impedance sensing device capable of validating anti-cancer drugs in a three-dimensional cancer cell model cultured within a microfluidic chamber. Specifically, a microgroove impedance sensor integrated in the microfluidic cell culture device measures electrical impedances, indicating the viability of three-dimensionally cultured cancer cells, so the device can identify cancer cell death by anti-cancer drugs.

In particular, studies linked to cell trapping techniques have actively progressed to acquire specific characteristics by positioning a single cell between electrodes to measure electrical impedance. These studies are detailed in the next chapter along with the introduction of cell-trapping techniques.

## 3. Various Technologies of Single-Cell Trapping on Microfluidic Impedance Biosensors

This section summarizes cell trapping techniques that are the basis of microfluidic impedance biosensors for analyzing single cells. Three latest technologies that can be efficiently analyzed in single cell units were selected and methods and principles were explained.

### 3.1. Dielectrophoretic (DEP)

Dielectrophoresis (DEP) is a powerful technique that can be used to control various biological cells, including cell sorting, cell transport, and cell trapping. DEP was defined by Herbert Pohl in 1951 and refers to a phenomenon in which a directional force is applied to a particle by an induced dipole when the particle is placed in a non-uniform electric field. In addition, because this technique can be applied to all polarizable particles, it can be utilized for moving, separating, and collecting various biological particles, including cells. The strength and direction of the net force applied to the particle depend on the relative polarization difference between the particle and the surrounding medium. When the polarity of the particle is greater than the polarity of the medium, the particle moves in a direction in which the electric field is relatively dense, which is called positive dielectrophoresis (p-DEP). Conversely, when the polarity of the medium surrounding the particle is greater than the polarity of the particle, the medium moves in a direction in which the electric field is relatively dense. As a result, the particle is pushed in the direction of the lower density of the electric field, which is called negative dielectrophoresis (n-DEP). The DEP force relies on several parameters such as the electrical and dielectric properties of the particle and medium, and the frequency of the applied AC electric field. Because the strength of the DEP force received by each cell is different at the same frequency of the AC electric field, it can be used to manipulate or separate different cell types. The total cell capacitance reflecting the plasma membrane region depends on physical characteristics such as cell size, wrinkles, and folds, as well as physiological conditions such as apoptosis [39]. It is possible to manipulate, and isolate, other cell types based on the difference in cell membrane capacitance by using DEP. These isolated cells were used for molecular analysis while maintaining a viable state.

Nguyen et al. detected lung circulating tumor cells (CTCs) by combining DEP manipulation with impedance measurement using circular microelectrodes within a single microfluidic device. Figure 2a shows a schematic diagram of the proposed device structure for measuring the impedance of the CTCs. The outside circles (blue electrodes) called manipulating electrodes were used for the DEP-based cell concentration [40]. Two pairs of central electrodes (red electrodes), called sensing electrodes, were used to pull, capture, and sense target cells. In the process of operation, cells were first trapped between the two largest circular electrodes due to positive DEP and hydrodynamic drag forces and when the electric field was switched to the inner circle, the cells moved to the center electrode.

Soo Hyeon Kim et al. presented a high-throughput single-cell array integrating the original acoustofluidic chip and an electroactive microwell array (EMA) to eliminate the bottleneck caused by cell trapping and subsequent single-cell analysis (Figure 2b) [41]. The acoustofluidic chip leads to an increase in the efficiency of the EMA through a constant outlet flow rate independent of the inlet flow rate. The human prostate cancer cell line DU145 was used to confirm this. The cell recovery, which is the percentage ratio of the number of captured cells to the number of introduced cells, was 96 ± 0.8% at an inlet flow rate of 20 μL min^−1^. The trapping efficiency of this system was increased by 4.2 times compared to the inherent trapping efficiency of conventional EMA. In addition, this system demonstrated a high trapping efficiency of 65 ± 13% even at a flow rate of 100 μL min^−1^.

**Figure 2 biosensors-11-00412-f002:**
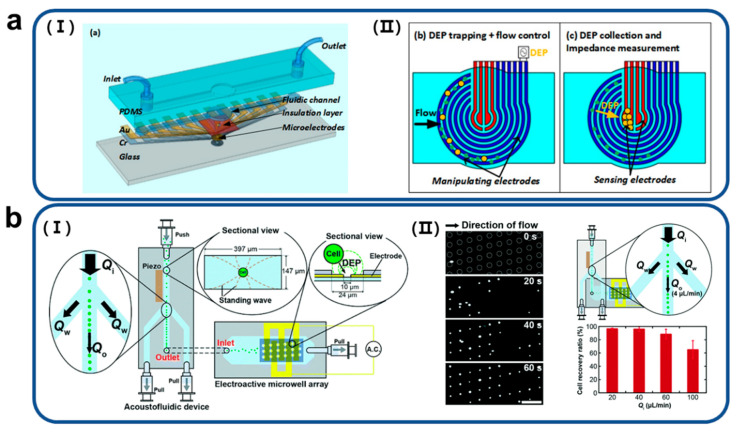
(**a**) Schematic diagram of (I) a microfluidic device combined with (II) circular microelectrodes capable of dielectrophoretic manipulation and electrical impedance measurement of target cells. Reproduced with permission from [40]. Copyright Lab on a Chip 2015. (**b**) (I) Schematic image of an acoustofluidic-based microfluidic device for high-throughput arrangement to the single-cell level. (II) Cells moving at a high-volume flow at the inlet of the microsystem were subsequently captured one by one in a microwell array through the dielectrophoresis principle. This device overcomes the bottleneck of cell analysis using a large sample volume. Reproduced from [41] with permission from the Royal Society of Chemistry.

### 3.2. Droplet

Droplet-based microfluidic technology is one of the primary analysis tools for single-cell research, offering precise individual manipulation of each droplet in space and time. Picoliter-to-sub microliter volumes of liquids work as individual sample containers or discrete volumes for performing isolated chemical or biological reactions. For example, droplet sequencing technology encapsulates single cells with uniquely barcoded microparticles, then lysates the cells to capture mRNA, and thousands of cellular transcripts are formed, and the cellular origin of each transcript can be investigated [42]. Thousands of cells can be processed quickly and without damage in a single cell unit, and sample consumption can be minimized. Droplet-based microfluidic analysis techniques include technical methods such as droplet generation, reactant treatment on the sample, and approaches for multi-reactor, long-term monitoring and assay. Here, we introduce approaches based on droplet-based microfluidic systems.

Droplet-based microfluidic systems have advantages of providing reliable manipulation and precise control of individual droplets in time and space. Babahosseini et al. proposed a microfluidic trap with a droplet chamber and lateral bypass channels combined with a microvalve that allows numerous droplets to be captured and merged throughout a wide range of discrete droplet sizes [43]. This integrated microfluidic platform offers isolation of cell encapsulation and enables combinations of individual droplets with desired biomaterials by the droplet merging process in the merging chamber. Thus, individual droplets were stored in a merging chamber and could be comprised of the desired number of cells or reactant.

The fully multi-step on-chip workflow for a single cell was suggested by Chung et al., who reported on on-chip droplet generation, sorting, and merging for quantitative cell mRNA detection, providing a rapid, robust, high-throughput assay. The authors designed a snowman-like droplet storage array for a fluidic system based on an on-chip sorting and merging platform. Droplets of 125 µM diameter were first generated and stored in a droplet storage array, and then smaller droplets with a diameter of 60 µM and containing single cell flow to be stored in the paired storage sites. To perform the reaction between two different-sized droplets, the paired droplets were merged by electrohydrodynamic force. By using two different-sized droplets and pairing-merging wells, the authors performed a quantitative on-chip multistep droplet-based single cell assay [44].

In the case of long-term monitoring-based cell assays, especially adherent cells, cells are incubated under specific conditions. However, conventional droplet-based fluidic systems can provide poor environmental survival conditions owing to the fluidic pressure or loss of cell anchoring from the oil coating. Therefore, Hassanzadeh-Barforoushi et al. introduced a semi-droplet concept to support the culture of both adherent and non-adherent cells [45]. By designing an array of hundreds of dispersed nanoliter-volume semi-droplets, spatial confinement of cell-containing liquid in the indexed trap with supporting cell incubation environment enables long-term cell assays. This method offers single-cell trapping and incubation while maintaining a chemically isolated indexed volume without microfluidic expertise.

### 3.3. Microstructures

Among single-cell capture methods, there is a microstructure-based single-cell trapping method, which shows high efficiency and accuracy. This method allows for the detection and quantification of specific cells at the single-cell level in complex solutions. For this reason, microstructure-based trapping has been actively applied to develop medical diagnoses and to analyze food safety. Techniques have been developed to use fluid flow and gravity in micro-scale structural arrays to capture and classify single cells within a certain space without damage [46]. Further development has taken place here to separate specific size, specific target cells, and a bandpass filter has been developed that can separate only cells of specific size from cells of several sizes using fluids [47]. In this section, cell trapping techniques based on microstructures such as microchannels or microwell arrays are described.

Mansor et al. investigated a microchannel and microneedles for measuring the impedance of cancer cells [48]. The microchannel was fabricated using PDMS as a low-cost soft lithography process. Specifically, this approach was applied to obtain information about cells passing through the sensing area using microchannels. Zhal et al. introduced a 3D microstructure for single-cell culture in a digital microfluidic system. The 3D microstructure was manufactured on a large scale by photolithography using an SU-8 negative resist (with a height of approximately 10µM and a distance of 300 µM) (Figure 3a) [49]. The system has a high single-cell trapping efficiency and can be used for cell analysis. D.-H. Lee et al. applied a microstructure array for size-selective single-cell trapping, which was established on a microfluidic platform as described in Figure 3b [50]. The developed platform analyzed fully integrated blood samples that enable size-selective cell separation and high-efficiency capture of single cells from diluted blood samples. The microfluidic platform sequentially filtered large and small cells using two different filters and successively separated medium-sized target cell populations. A target cell line, a medium-sized target cell population, was isolated and trapped without external force. This study presents a size-selective single-cell analysis platform for medical diagnosis. In addition, D.-H. Lee et al. used microstructure to trap single leukocytes and single leukemia cells without labeling in a complex sample, blood, and distinguished each cell using phasor-fluorescence lifetime microscopy [51].

## 4. Single-Cell Analysis of Microfluidic Impedance Biosensors

High throughput analysis capabilities for the biological properties of Impedance biosensors introduced in the previous section. It was suggested that a single cell can be accurately trapped at a desired location in an impedance microfluidic chip using DEP, droplet, and microstructures structures. These technologies have potential to accurately analyze the geometry of single cell units and cell activity. This section examines the applications of these combined technologies and summarizes what characteristics of cells have been identified. Although only some of the studies that have been conducted have been presented, there is a possibility that these studies will proceed further in the future. In addition to this, the contents of studies that analyzed single cells were added. Although these studies did not trap single cells by the method presented in the previous section, they were able to obtain very useful biological information obtained by analyzing various types of single cells. Through this, it is expected that basic but important features can be obtained for single cell analysis in the future with the advantages of single cell trapping techniques presented in the previous section, so the contents are summarized.

HeLa cells are transported by the principle of liquid dielectrophoresis (LDEP) and are trapped in the SU-8 cavity by DEP [52]. This study developed an optimized microstructure that reduced the complex process for single-cell measurement and increased efficiency to trap single cells in precise locations. In addition, a droplet microfluidic microelectrode structure was developed to monitor the osteogenic differentiation of single bone marrow mesenchymal stem cells over time [53]. The possibility of long-term monitoring of electrical properties according to biological changes in a single cell unit was suggested.

Electrical impedance analysis based on dielectric properties, cell size and composition of sickle RBCs was performed using an effective non-invasive and label-free microfluidic device at the single-cell scale (Figure 4a) [54]. The impedance properties of one healthy and three sickle samples were measured at frequencies of 156 kHz, 500 kHz, and 3 MHz, respectively, under normoxic and hypoxic conditions to discriminate between these two types of samples. The measured Δ|Z| of normal RBCs (normoxia) have a value of 2.4 × 10^7^ ± 0.75 × 10^7^ Ω, much higher than a value of 1.1 × 10^7^ ± 0.45 × 10^7^ Ω of sickle cells (hypoxia) at a relatively low frequency of 156 kHz and corresponding values of Δθ were −0.90 ± 0.02 rad and −0.59 ± 0.31 rad for normal and sickle cells, respectively. These results suggest that the impedance monitoring tool developed in this study under normoxia suggests that sickle cells can be distinguished from normal cells. Furthermore, the results suggest that differences in the electrical impedance signals of sickle cells measured under hypoxic and normoxic conditions could provide additional information indicative of cellular sickle events. These findings lead to the conclusion that electrical impedance differences may serve as novel biomarkers for sickle cell disease.

Cells were analyzed by microfluidic impedance flow cytometry, and a remarkable double-peak signal response was revealed by changes in cell membrane capacitance [55]. This peak signal was limited to lower frequencies (400−800 kHz) in the β-dispersion region. This was used to accurately differentiate between normal and glutaraldehyde-treated RBC populations based on changes in cell membrane capacitance. A double peak signal was used to identify cell populations within the PBMC mixture. This allows the measurement of both cell size and cell membrane properties at a single frequency, which greatly simplifies the system and reduces costs and improves existing approaches. In particular, a double-peak profile was clearly observed in lymphocytes and monocytes. Two subpopulations of lymphocytes were identified based on their peak ratios. In total, three distinct cell subpopulations were identified. Through the peak ratio, changes in the cell membrane dielectric properties can be captured and analyzed simultaneously and independently at a single frequency.

**Figure 4 biosensors-11-00412-f004:**
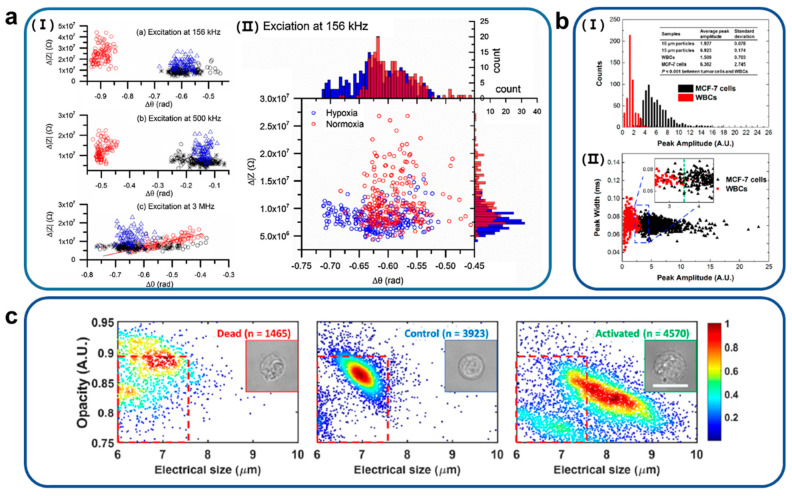
(**a**) (I) Differential impedance scatter plots that present discrimination of sickle RBCs and normal RBCs under normoxia at different frequencies. (II) Scatter values of Δ|Z| of sickle RBCs under normoxia and hypoxia at electrical frequency 156 kHz. Reproduced with permission from [54]. Copyright Sensors and Actuator B: Chemical 2018. (**b**) (I) Distribution of impedance peak amplitudes versus counts of MCF-7 cells and white blood cells (WBCs) and (II) scatterplots that explain correlation between peak amplitudes versus peak width of MCF-7 cells and WBCs. Reproduced with permission from [56]. Copyright American Chemical Society 2017. (**c**) Profile of opacity versus electrical size of dead lymphocytes, normal lymphocytes and activated lymphocytes. Profiling opacity versus biophysical size that presents a characterization of peripheral blood mononuclear cells (PBMCs) treated with phytohemagglutinin (PHA). Reproduced with permission from [57]. Copyright Sensors and Actuators B: Chemical 2021.

To distinguish MCF-7 cells from white blood cells (WBCs), liquid electrodes with *Ag*/*AgC**l* wires filled with a highly conductive electrolyte were used (Figure 4b) [56]. These authors established a linear relationship between the amplitude of the impedance and the volume of the particles and calculated the directness of the cells, assuming that the cells were spherical. As a result, 99% of the WBCs were included below the peak amplitude of 3.5, but more than 96% of the WBCs were higher than this value. The authors developed an efficient liquid electrode-based impedance micrometer with a high throughput of 5000 cells/s, as well as being able to accurately identify cells without labeling.

A novel impedance-based microfluidic technique was presented to analyze biophysical responses of antigen-specific T lymphocytes without labeling (Figure 4c) [57]. A spiral inertial microfluidic cell classifier was developed, which eliminated small inactive lymphocytes and collected activated lymphocytes at substantial success rates using the principle of hydrodynamic-based single-stream particle focusing. Differences were observed in membrane electrical impedance of in not only cell size but also dead lymphocytes, healthy lymphocytes, and activated (*CD3*/*CD28*) lymphocytes.

## 5. Conclusions

This review summarizes the fundamental principles of electrical impedance biosensors, various applications for important target analytes, efficient methods to manipulate single cells, and explanations of single-cell characteristics extracted from these biosensors. Impedance biosensors, combined with technologies to trap single cells such as the DEP system, droplets flow manipulation, and microstructure fabrications, can accurately describe the micromorphological and pharmacological functions of single targets with high-throughput measurement. Useful information from living single cells can be obtained while maintaining high survival rates for cells through fine adjustment of fluid flow in a non-invasive way and can lead to building a basic understanding in a variety. In addition, it was also able to analyze diverse targets such as DNA hybridization, human serum albumin, bacteria, and cancer cells, and are considered valuable research technologies. These technologies can be utilized as advanced analytical models because they are suitable for analyzing real-time events while exhibiting improved sensitivity and selectivity. These technologies require relatively less workforce or demand for micro-scale structures manufactured by soft lithography than nanostructures based on nanotechnology. By combining the SPR phenomenon to amplify signals for biomolecules, the sensitivity and selectivity of the sensor can be improved in the intended direction. In summary, electrical impedance biosensors with microfluidic chips for single-cell trapping have great potential for application in significant areas such as disease diagnosis, cell micro-reaction control and screening.

## Figures and Tables

**Figure 3 biosensors-11-00412-f003:**
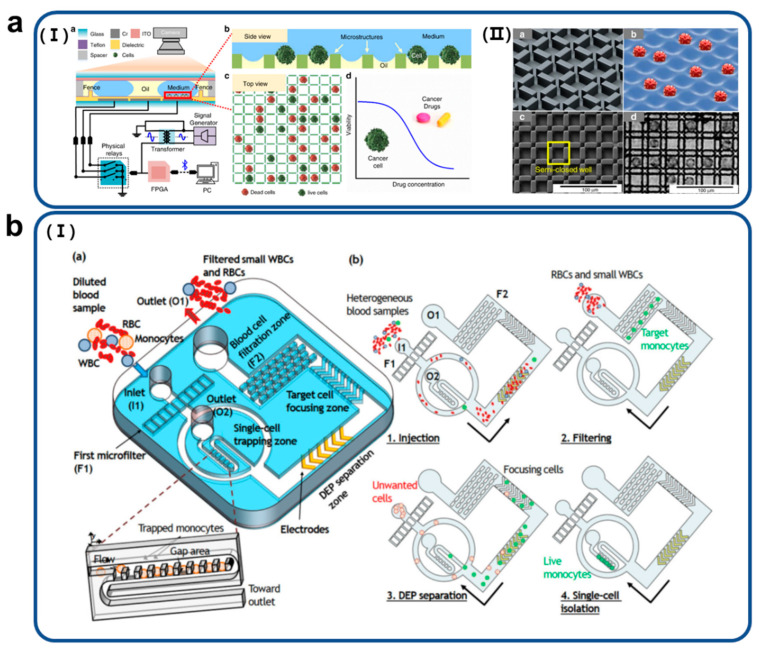
(**a**) (I) Schematic images (side view) of developed digital microfluidic device and (II) 3D microstructures to isolate single cells and investigate the cell response for drug sensitivity test. Reproduced with permission from [48]. Copyright Microsystems & Nanoengineering 2020. (**b**) (I) The integrated microfluidic platform enabling size-selective single-cell isolation, composed of hydrodynamic microfiltration, dielectrophoretic separation, and a microwell array. Reproduced with permission from [49]. Copyright Biomicrofluidics 2021.

## Data Availability

Data sharing not applicable.

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
