# Peer review of "A Review of Advanced Impedance Biosensors with Microfluidic Chips for Single-Cell Analysis"

_biosensors, 2021, doi:10.3390/bios11110412_

Round 1
Reviewer 1 Report
In this review paper, the author has introduced the microfluidic-based EIS biosensor for target analysis. Some questions need to be addressed.
- The topic needs to change. Since the author also introduced a DNA-DNA sensor, the single-cell analysis is misleading.
- It is tough to say which part is the principal part, microfluidic device or EIS technique? Simply putting these two together may make the reader getting confusing. I suggest the author focusing on one topic instead of piling up these two.
- What is the meaning of each component in the equation? Without any explanation, it may be better to remove the equation.
- The author mentioned the difference in specificity or sensitivity of impedance sensors compared with the PCR technique. What may be the reason?
Reviewer 2 Report
The authors try to cover in an ambitious review paper several different topics (i.e. small molecule sensing, cell trapping, and single-cell sensing) which are quite broad, and recently deemed worthy each of their own dedicated review paper (see e.g. Chircov et al [Molecules 2020], Petchakup et al [Micromachiness 2017], Deng et al [Processes 2020], Zhu et al [Talanta 2021], Hiramoto et al [Front Chem. 2019]). The authors didn’t provide a convincing connection among examined topics, for example reviewing systems using both electrical and optical (e.g. ref 45-46) readouts, and batch and continuous-flow (e.g. ref 47, 48, 50) systems without an apparent common thread.
Moreover, each topic is not exhaustively examined. For example, several common methods for cell trapping are missing, including acustophoresis (e.g. Yin et al [Colloids Surf. B Biointerfaces 2017]), magnetophoresis (e.g. Scherr et al [Biomicrofluidics 2016]), and optical tweezers (e.g. Li et al [Opt. Express 2018]), as well as several important studies relevant to the microfluidic impedance cytometry field (e.g. Spencer et al [Biomicrofluidics 2014] and Honrado et al [Analytical and Bioanalytical Chemistry 2020]).
Additionally, authors provide little critical analysis of existing literature, and mostly provide a list of results from previous papers. They don’t provide a clear description of the state-of-the-art, of problems overcome by each paper, remaining challenges, and/or future perspectives.
As a result, this review paper feels dispersive and incomplete.
Moreover, several other minor issues remain, including:
- Writing could be improved throughout the paper to improve readability.
- There are several typos throughout the paper that should be corrected.
- Figure 3c is cited (line 298) but not present in text.
- Section 4 is titled “Types of microfluidic devices for single cell trapping” but mostly describes continuous flow systems where cells were not trapped.
- Figure 1a and Figure 3bI are of poor quality and unreadable.
Author Response
Reviewer #2
Comments to the Author
The authors try to cover in an ambitious review paper several different topics (i.e. small molecule sensing, cell trapping, and single-cell sensing) which are quite broad, and recently deemed worthy each of their own dedicated review paper (see e.g. Chircov et al [Molecules 2020], Petchakup et al [Micromachiness 2017], Deng et al [Processes 2020], Zhu et al [Talanta 2021], Hiramoto et al [Front Chem. 2019]). The authors didn’t provide a convincing connection among examined topics, for example reviewing systems using both electrical and optical (e.g. ref 45-46) readouts, and batch and continuous-flow (e.g. ref 47, 48, 50) systems without an apparent common thread.
Moreover, each topic is not exhaustively examined. For example, several common methods for cell trapping are missing, including acustophoresis (e.g. Yin et al [Colloids Surf. B Biointerfaces 2017]), magnetophoresis (e.g. Scherr et al [Biomicrofluidics 2016]), and optical tweezers (e.g. Li et al [Opt. Express 2018]), as well as several important studies relevant to the microfluidic impedance cytometry field (e.g. Spencer et al [Biomicrofluidics 2014] and Honrado et al [Analytical and Bioanalytical Chemistry 2020]).
Additionally, authors provide little critical analysis of existing literature, and mostly provide a list of results from previous papers. They don’t provide a clear description of the state-of-the-art, of problems overcome by each paper, remaining challenges, and/or future perspectives.
As a result, this review paper feels dispersive and incomplete.
Moreover, several other minor issues remain, including:
- Writing could be improved throughout the paper to improve readability.
- There are several typos throughout the paper that should be corrected.
- Figure 3c is cited (line 298) but not present in text.
- Section 4 is titled “Types of microfluidic devices for single cell trapping” but mostly describes continuous flow systems where cells were not trapped.
- Figure 1a and Figure 3bI are of poor quality and unreadable.
Our response:
We appreciate the reviewer’s helpful comment to provide various advice to improve the content of this review. The opinions of other reviewers also had similar opinions and we suggested further improvement about importance of cell trapping technologies.
- According to the advice that the topics look diverse, comments were added to the introduction as follows so that the order of reviews could be understood.
[Page 2 (line 75-86)]
In this article, we explored the concept of and recent studies on microfluidic impedance single-cell biosensors, a device that combines an electrical impedance biosensor and a microfluidic device to perform analysis on a single cell. First, we explored the definition, principle, and application of impedance microfluidic sensors in various biological applications to inform useful technology to be used as a tool for single-cell analysis. Second, concepts and recent studies of microfluidic devices that can classify single cells and move them to specific locations for impedance single-cell biosensing were also investigated and presented. Finally, we sequentially analyzed recent research on microfluidic impedance biosensors for single cells. The results of the investigation and analysis of microfluidic impedance sensors for single-cell analysis are expected to present a milestone in the development of advanced biosensing systems that analyze the characteristics of single cells using microfluidic impedance sensing.
- The advantages and effectiveness of the measurement systems with proposed technologies for cell trapping were additionally mentioned along with cell viability and classification. Measured physical characteristics of the cells were further added to explain cellular mechanisms depended on cell state and external environment.
[Page 7 (Line 261-266)]
The total cell capacitance reflecting the plasma membrane region depends on physical characteristics such as cell size, wrinkles, and folds, as well as physiological conditions such as apoptosis [39]. It is possible to manipulate, and isolate other cell types based on the difference in cell membrane capacitance by using DEP. These isolated cells were used for molecular analysis while maintaining a viable state.
[Page 9 (line 344-350)]
In this section, cell trapping techniques based on microstructures such as microchannels or microwell arrays are described. Techniques have been developed to use fluid flow and gravity in micro-scale structural arrays to capture and classify single cells within a certain space without damage [46]. Further development has taken place here to separate specific size, specific target cells, and a bandpass filter has been developed that can separate only cells of specific size from cells of several sizes using fluids [47]. In this section, cell trapping techniques based on microstructures such as microchannels or microwell arrays are described.
- In the section where the importance of technology was insufficient, related content was additionally written. Thank you for mentioning important parts.
[Page 7 (line 291-296)]
For example, droplet sequencing technology encapsulates single cells with uniquely barcoded microparticles, then lysates the cells to capture mRNA, and thousands of cellular transcripts are formed, and the cellular origin of each transcript can be investigated [42]. Thousands of cells can be processed quickly and without damage in a single cell unit, and sample consumption can be minimized.
- The letter 'Figure 3c' was incorrectly entered and deleted.
[Page 8 (Line 322-324)]
Therefore, Hassanzadeh-Barforoushi et al. introduced a semi-droplet concept to support the culture of both adherent and non-adherent cells (Figure 3c) [43].
- The incorrectly written title was accurately corrected.
[Page 11 (Line 372)]
4. Types of microfluidic devices for single cell trapping
4. Single-cell analysis of microfluidic impedance biosensors
- The readability of Figures 1a and 3b was enhanced.

Reviewer 3 Report
Line 119: "A general method for detecting a specific virus is to culture the virus in specimens and analyze them by real-time polymerase chain reaction (PCR)." – Culturing virus in host cells is not generally necessary for viral detection, as PCR detection of specific nucleic acid sequences is quite sensitive and specific. The value of microfluidic devices over PCR is generally the following: speed of detection, use in resource limited settings, and potential ease of use (lab on a chip). Detection of proteins associated with certain viruses is generally much faster and performed using ELISA (as you mention on line 146). The benefits of using microfluidic devices instead of ELISA should be discussed: the analyte is put in much closer proximity to the detector due to the confined space leading to the antibody diffusing to the protein much more quickly (leading to faster detection and use of less expensive antibody reagents), electrical signal detection can be more sensitive than optical detection (greater sensitivity).
Line 126: “Compared to the virus detection method using real-time reverse transcriptase PCR, the microfluidic impedance sensor provided 100 % sensitivity and 64 % specificity” – What was the sensitivity and specificity of the RT-qPCR test for comparison? There are a series of detection limits on this page listed, but without any context as to how they are improved compared with conventional detection techniques. Do they have greater sensitivity or specificity? By how much? If they are not better in sensitivity or specificity compared with current lab diagnostic techniques, how are they better? Faster detection rates? By how much? Portability or use in resource-limited settings? Please elaborate.
Line 240: “The DEP force relies on several parameters such as the electrical and dielectric properties of the particle” – These inherent electrical and dielectric properties of cells (or particles) are directly linked to physical properties of the cell – including size, membrane composition, and interior complexity. It would be good to mention how these physical properties can be used to distinguish different cell types, or characterize a cell type.
DEP can also be used in a continuous flow to either attract or deflect cells to a sidewall of a channel, allowing for separation. Mention and description of such a device is important for this section, please consider adding.
Section 3.2: Droplet – you need to mention Drop-Seq, developed by the McCarroll lab http://mccarrolllab.org/dropseq/ https://www.illumina.com/science/sequencing-method-explorer/kits-and-arrays/drop-seq.html , as it is one of the most important and widely used methods of droplet-based single cell analysis. In this method, a barcoded bead is added to a single cell in a droplet, which is then lysed and the RNA is bound to the bead (RNA barcode), which is then sequenced. This allows for analysis of tissue-level heterogeneity of cells which would normally be lost when analyzing RNA from homogenized tissue samples, for discovery of new cell types.
Section 3.3: Microstructures – you should mention deterministic lateral displacement, which separates particles in continuous flow device using posts, and is based on primarily in differences in cell size (inertia), and cell deformity
Section 4. Types of microfluidic devices for single cell trapping – This title is not appropriate for this technique. Please describe impedance cytometry technique – how it works and what it is measuring. Impedance cytometry does not trap cells; it uses continuous flow to focus cells that are passing one at a time through a microchannel, which are then analyzed by changes to electrical impedance. What physical properties of the cell are being measured (at different applied frequencies) to characterize these cells?
Author Response
Reviewer #3
Comments to the Author
1) Line 119: "A general method for detecting a specific virus is to culture the virus in specimens and analyze them by real-time polymerase chain reaction (PCR)." – Culturing virus in host cells is not generally necessary for viral detection, as PCR detection of specific nucleic acid sequences is quite sensitive and specific. The value of microfluidic devices over PCR is generally the following: speed of detection, use in resource limited settings, and potential ease of use (lab on a chip). Detection of proteins associated with certain viruses is generally much faster and performed using ELISA (as you mention on line 146). The benefits of using microfluidic devices instead of ELISA should be discussed: the analyte is put in much closer proximity to the detector due to the confined space leading to the antibody diffusing to the protein much more quickly (leading to faster detection and use of less expensive antibody reagents), electrical signal detection can be more sensitive than optical detection (greater sensitivity).
Our response:
We appreciate the reviewer’s helpful comment to improve content of the manuscript. According to the reviewer's comment, we revised and added sentences as below to provide more accurate information to readers.
[Page 4 (line 113-115)]
A general method for detecting a specific virus is to culture the virus in specimens and analyze them by real-time polymerase chain reaction (PCR). This method is highly accurate but takes a long time for the diagnostic results to be obtained.
[Page 4 (line 154-167)]
The microfluidic impedance HIV-1 biosensing device developed by H. Shafiee et al. presented the possibility of providing virus detection and point-of-care (POC) diagnosis more quickly than conventional analytical methods such as enzyme-linked immunosorbent assay (ELISA). To make this technique more suitable for use in POC di-agnostics, X. Li et al. fabricated working electrodes coated with carbon ink, zinc oxide (ZnO) nanowires, and antibodies that can be attached to p24 antigen, one of the HIV’s biomarkers on a paper fluidic substrate [28]. They applied the electrodes to high-sensitive HIV detection using electrical impedance spectroscopy (A limit-of-detection of p24 antigen = 0.4 pg/mL). Also, this device and detection technique demonstrated the potential to be utilized for rapid diagnosis of COVID-19 by serological analysis. Compared to ELISA, microfluidic impedance sensor to detect virus enable faster detection and the use of low-cost antibody reagents since analytes are placed much closer to the detector. Also, electrical signal detection in the microfluidic impedance sensor can be more sensitive than optical measurement in ELISA.
2) Line 126: “Compared to the virus detection method using real-time reverse transcriptase PCR, the microfluidic impedance sensor provided 100 % sensitivity and 64 % specificity” – What was the sensitivity and specificity of the RT-qPCR test for comparison? There are a series of detection limits on this page listed, but without any context as to how they are improved compared with conventional detection techniques. Do they have greater sensitivity or specificity? By how much? If they are not better in sensitivity or specificity compared with current lab diagnostic techniques, how are they better? Faster detection rates? By how much? Portability or use in resource-limited settings? Please elaborate.
Our response:
We appreciate the reviewer’s helpful comment to improve the manuscript and the degree of readers’ understanding. In a reference article (R. Wang, et al. J. Virol. Methods (2011)) presented in this manuscript, a sensitivity was defined as a ratio of a number of samples identified as positive by both impedance sensor/real-time reverse transcriptase polymerase chain reaction (PCR) and virus culture (NTP) to a sum of NTP and a number of samples identified as negative by impedance sensor/real-time reverse transcriptase PCR and positive by virus culture (NFN). Also, a specificity was calculated as a ratio of a number of samples identified as negative by both impedance sensor/real-time reverse transcriptase PCR and virus culture (NTN) to a sum of NTN and a number of samples identified as positive by impedance sensor/real-time reverse transcriptase PCR and negative by virus culture (NFP).
Definitions of sensitivity and specificity were added to the manuscript as follows. In addition, we confirmed that some sentences were misrepresented in the numerical comparison, so we corrected them as suggested as follows. Also, we added the advantages of this microfluidic impedance virus sensor to the end of the description in the manuscript. We appreciate reviewer’s correction request of content that could confuse readers.
[Page 4 (line 113-138)]
A general method for detecting a specific virus is to culture the virus in specimens and analyze them by real-time polymerase chain reaction (PCR). This method is highly accurate but takes a long time for the diagnostic results to be obtained. Microfluidic impedance virus sensors have been actively studied because they provide the detection of selected viruses, which are simpler and faster than the conventional method. R. Wang et al. developed a technique that can test avian influenza virus H5N2 with high sensitivity using a portable impedance sensor, which consists of 25 pairs of microelectrodes and a microfluidic channel, and antibody-coated magnetic nanobeads [24]. For performance comparison between the microfluidic impedance sensor and real-time reverse transcriptase PCR, there were two parameters, a sensitivity and specificity, used as performance indicators. A sensitivity was defined as a ratio of a number of samples identified as positive by both a virus analytical method (impedance sensor/real-time reverse transcriptase PCR) and virus culture (NTP) to a sum of NTP and a number of samples identified as negative by the analytical method and positive by virus culture (NFN). Also, a specificity was calculated as a ratio of a number of samples identified as negative by both the analytical method and virus culture (NTN) to a sum of NTN and a number of samples identified as positive by the analytical method and negative by virus culture (NFP). For viral samples extracted from the trachea of infected or normal chickens, the developed microfluidic impedance sensor provided 100 % sensitivity (equal to the sensitivity of real-time reverse transcriptase PCR) and 64 % specificity (less than the specificity (= 100 %) of real-time reversed transcriptase PCR). In the case of samples extracted from cloacal swabs of infected or normal chickens, the microfluidic impedance sensor had an ideal specificity (100 %), which was higher than it (= 69 %) of real-time reversed transcriptase PCR, and lower sensitivity (55 %) than real-time reversed transcriptase PCR-based virus detection (81 %). Compared to PCR, the developed microfluidic impedance virus sensor has advantages of being small in size (portable), having shorter measurement time (30 to 60 minutes), and being easier to apply.
3) Line 240: “The DEP force relies on several parameters such as the electrical and dielectric properties of the particle” – These inherent electrical and dielectric properties of cells (or particles) are directly linked to physical properties of the cell – including size, membrane composition, and interior complexity. It would be good to mention how these physical properties can be used to distinguish different cell types, or characterize a cell type.
DEP can also be used in a continuous flow to either attract or deflect cells to a sidewall of a channel, allowing for separation. Mention and description of such a device is important for this section, please consider adding.
Our response:
We appreciate the reviewer’s helpful comment regarding additional information on what physical properties were used to distinguish and analyze cell types, and how important DEP is in measuring these properties. We have additionally written the following sentences to conform to this advice.
[Page 7 (Line 261-266)]
The total cell capacitance reflecting the plasma membrane region depends on physical characteristics such as cell size, wrinkles, and folds, as well as physiological conditions such as apoptosis [39]. It is possible to manipulate, and isolate other cell types based on the difference in cell membrane capacitance by using DEP. These isolated cells were used for molecular analysis while maintaining a viable state.
4) Section 3.2: Droplet – you need to mention Drop-Seq, developed by the McCarroll lab http://mccarrolllab.org/dropseq/, https://www.illumina.com/science/sequencing-method-explorer/kits-and-arrays/drop-seq.html , as it is one of the most important and widely used methods of droplet-based single cell analysis. In this method, a barcoded bead is added to a single cell in a droplet, which is then lysed and the RNA is bound to the bead (RNA barcode), which is then sequenced. This allows for analysis of tissue-level heterogeneity of cells which would normally be lost when analyzing RNA from homogenized tissue samples, for discovery of new cell types.
Our response:
We appreciate the reviewer’s helpful comment regarding the very important supplementary points in the part described in droplet technology. We have added the following sentences to this paper, along with references. What we emphasized is that single-cell microfluidic analysis is fast, simple, and high-throughput analysis. If we add the points mentioned in the review, it is expected that the importance and high utility of the droplet technology will be clearly explained.
[Page 7 (line 291-299)]
For example, droplet sequencing technology encapsulates single cells with uniquely barcoded microparticles, then lysates the cells to capture mRNA, and thousands of cellular transcripts are formed, and the cellular origin of each transcript can be investigated [42]. Thousands of cells can be processed quickly and without damage in a single cell unit, and sample consumption can be minimized.
5) Section 3.3: Microstructures – you should mention deterministic lateral displacement, which separates particles in continuous flow device using posts, and is based on primarily in differences in cell size (inertia), and cell deformity.
Our response:
We appreciate the reviewer’s helpful comment regarding the additional mention of later disposition. We found more important papers related to the advice and added the contents as follows. Thank you for making up for the shortcomings.
[Page 9 (line 344-350)]
In this section, cell trapping techniques based on microstructures such as microchannels or microwell arrays are described. Techniques have been developed to use fluid flow and gravity in micro-scale structural arrays to capture and classify single cells within a certain space without damage [46]. Further development has taken place here to separate specific size, specific target cells, and a bandpass filter has been developed that can separate only cells of specific size from cells of several sizes using fluids [47]. In this section, cell trapping techniques based on microstructures such as microchannels or microwell ar-rays are described.
6) Section 4. Types of microfluidic devices for single cell trapping – This title is not appropriate for this technique. Please describe impedance cytometry technique – how it works and what it is measuring. Impedance cytometry does not trap cells; it uses continuous flow to focus cells that are passing one at a time through a microchannel, which are then analyzed by changes to electrical impedance. What physical properties of the cell are being measured (at different applied frequencies) to characterize these cells?
Our response:
We appreciate the reviewer’s helpful comment regarding pointing out the inadequacy of the title and additional points on impedance cytometry technology.
This wrong title seems to have made it difficult to understand the content. 1) The wrong title was corrected. 2) In addition, a total of four research results are summarized in this section, and the part pointed out in the advice already explains which parameters were measured for each study and which cell characteristics were analyzed in the sentence below. 3) However, this explanation was further explained in the missing part.
1) A correction
[Page 11 (Line 372)]
4. Types of microfluidic devices for single cell trapping
4. Single-cell analysis of microfluidic impedance biosensors
2) The parts already mentioned.
[Page 11 Line (412-414)] This allows the measurement of both cell size and cell membrane properties at a single frequency, which greatly simplifies the system and reduces costs and improves existing approaches.
[Page 11 Line (422-424)] These authors established a linear relationship between the amplitude of the impedance and the volume of the particles and calculated the directness of the cells, assuming that the cells were spherical.
[Page 11 Line (432-434)] Differences were observed in membrane electrical impedance of in not only cell size but also dead lymphocytes, healthy lymphocytes, and activated (CD3/CD28) lymphocytes.
3) An addition
[Page 11 (Line 392-393)]
Electrical impedance analysis based on dielectric properties, cell size and composition of sickle RBCs was performed using an effective non-invasive and label-free microfluidic device at the single-cell scale (Figure 4a) [47].

Round 2
Reviewer 1 Report
No further questions.
Reviewer 2 Report
In my opinion, the authors failed to address the main issue of the manuscript, which is the lack of a common thread connecting the reviewed literature.
After the initial introduction [Section 1], the authors have structured their manuscript as follows:
- [Section 2] They analyze some broad applications of impedance biosensors (including measurement of DNA, viruses, and bacteria);
- [Section 3] They analyze a few types of microfluidic systems for single cell trapping;
- [Section 4] They analyze some microfluidic systems for impedance-based single cell analysis.
As a consequence, the reader would expect to find in this review many papers exploiting the combination of single cell trapping and impedance measurement. This intention is also shared by the authors, as highlighted by the paragraph at line 388 of the revised manuscript:
“If high throughput analysis capabilities for the biological properties of Impedance biosensors introduced in the previous section, and single cell trapping techniques that preserve the state of cells are combined, it has potential to accurately analyze the geometry of single cell units and cell activity. This section examines the applications of these fusion technologies and summarizes what characteristics of cells have been identified.”
Nevertheless, only 1 work described in Sections 2-4 falls into this category (Ref. 40 of the revised manuscript). All other reviewed work, either reports systems for single cell trapping or for single cell impedance analysis, but no system combining both aspects. In particular, Ref. 41, 42, 44-50 describe systems for cell trapping with optical readouts (NOT impedance-based), whereas Ref. 51-54 describe systems for single cell impedance analysis under flow conditions (where cells are NOT trapped).
As a result, each different section seems disconnected from the others, and too superficial to be worthwhile on its own.
Additional remark:
The authors refer to a paper in the text at line 352 of the revised manuscript, but no citation is provided:
“Mansor et al. investigated a microchannel and microneedles for measuring the impedance of cancer cells. The microchannel was fabricated using PDMS as a low-cost soft lithography process. Specifically, this approach was applied to obtain information about cells passing through the sensing area using microchannels.”
I assumed the authors were referring to Appl. Sci. 2017, 7(2), 170, which describes another system for impedance‑based flow cytometry, where cells are not trapped.
